# A Schrödinger Equation for Evolutionary Dynamics

**Vi D. Ao [1], Duy V. Tran [2,†], Kien T. Pham [3,†] , Duc M. Nguyen [4,†], Huy D. Tran [5], Tuan K. Do [6], Van H. Do [7] and Trung V. Phan [8,*]**

[1] Department of Physics, VNUHCM University of Science, 227 Nguyen Van Cu, Ho Chi Minh 700000, Vietnam; 22130216@student.hcmus.edu.vn

[2] Department of Mechanical Engineering, VNUHCM University of Technology, 226 Ly Thuong Kiet, Ho Chi Minh 110000, Vietnam

[3] Department of Aerospace Engineering, School of Transportation Engineering, Hanoi University of Science and Technology, 01 Dai Co Viet, Hanoi 100000, Vietnam; ubkpk@student.kit.edu

[4] Department of Mathematics, University of Chicago, 5801 S Ellis Ave, Chicago, IL 60637, USA; ducnguyenmanh@uchicago.edu

[5] Department of Physics, The Hong Kong University of Science and Technology, Clear Water Bay, Kowloon, Hong Kong, China

[6] Department of Mathematics, University of California, Los Angeles (UCLA), 520 Portola Plaza, Los Angeles, CA 90095, USA

[7] Homer L. Dodge Department of Physics and Astronomy, University of Oklahoma, 440 W. Brooks St., Norman, OK 73019, USA

[8] Department of Molecular, Cellular and Developmental Biology, Yale University, 260 Whitney Ave, New Haven, CT 06511, USA

* Correspondence: tphan23@jh.edu
† These authors contributed equally to this work.

**Abstract:** We establish an analogy between the Fokker–Planck equation describing evolutionary landscape dynamics and the Schrödinger equation which characterizes quantum mechanical particles, showing that a population with multiple genetic traits evolves analogously to a wavefunction under a multi-dimensional energy potential in imaginary time. Furthermore, we discover within this analogy that the stationary population distribution on the landscape corresponds exactly to the ground-state wavefunction. This mathematical equivalence grants entry to a wide range of analytical tools developed by the quantum mechanics community, such as the Rayleigh–Ritz variational method and the Rayleigh–Schrödinger perturbation theory, allowing us not only the conduct of reasonable quantitative assessments but also exploration of fundamental biological inquiries. We demonstrate the effectiveness of these tools by estimating the population success on landscapes where precise answers are elusive, and unveiling the ecological consequences of stress-induced mutagenesis—a prevalent evolutionary mechanism in pathogenic and neoplastic systems. We show that, even in an unchanging environment, a sharp mutational burst resulting from stress can always be advantageous, while a gradual increase only enhances population size when the number of relevant evolving traits is limited. Our interdisciplinary approach offers novel insights, opening up new avenues for deeper understanding and predictive capability regarding the complex dynamics of evolving populations.

**Keywords:** Schrödinger equation; Rayleigh–Ritz variational method; Rayleigh–Schrödinger perturbation theory; fitness landscape; stress-induced mutagenesis; population dynamics

## 1. Introduction

Evolution is the primary driving force behind the diversity and complexity of life on Earth for billions of years [1,2], allowing for organism change and adaptation over time [3,4]. It emerges through the synergy between natural selection and genetic mutations, in which natural selection favors combinations of traits that enhance fitness [5] while genetic mutations introduce genetic variations that facilitate the emergence of new advantageous traits [6]. Within populations, ecological factors such as niche constraint [7–9] and

environmental stress [10,11] can exert their influence on these processes [12], even molding the trajectory and tempo of evolution [13].

The complex dynamics of evolving population can be captured by a Fokker–Planck equation on the evolutionary landscape [14], an abstract space of all possible genetic variations and their corresponding biological properties within a given ecological context [15]. The number of relevant evolving genetic traits corresponds to the dimensionality of this space [16], where every combination corresponds to an unique position. On the landscape, together with the ecological influence, we represent the mutation process with an effective diffusion [17] and the natural selection pressure with a fitness potential [18]. We show that there is an analogy between this formulation of evolutionary dynamics and the Schrödinger description for quantum mechanical particles [19], in which the manner of population evolution on the multi-dimensional landscape is almost similar to how a wavefunction behaves under a multi-dimensional energy potential in imaginary time [20]. Following this observation, we further discover that the stationary population distribution on the landscape corresponds exactly to the quantum ground-state wavefunction [19,21]. Such curious connection enables us to utilize various quantitative tools borrowed directly from quantum mechanics literature to quickly extract information about the steady population state of complex landscapes, even ones that lack exact analytical comprehension. For other examples of classical-quantum analogies where insights from quantum physics can illustrate classical phenomenon; see [22–26].

Understanding the consequences of evolution has always been a cornerstone of biological research [27], serving as a fundamental pursuit aimed at unravelling the possible outcomes arising from this transformative force, and shedding light on the foundational principles that shape and govern all life on Earth. There exist many distinct evolutionary regimes [28]. For pathogenic and neoplastic populations such as microbial organisms and cancer cells, stress-induced mutagenesis [10,11], in which mutational increase can be triggered due to high biological stress, assumes a prominent role. We consider *E.coli* bacteria after experiencing exposure to antibiotics [29,30]; they undergo elongation and stop dividing [31]. At the same time, inside the bacterium, the SOS response switches on, leading to the induction of low-fidelity error-prone replication polymerases and, consequently, there is a sharp increase in the mutation rate during DNA replication from the typically low value of $D_l \propto 10^{-9}$ to a high rate of $D_h \propto 10^{-5}$ mutations per base pair per generation [32,33]. There are also several other mechanisms by which genetic change can occur when organisms are under stress [34]. Laboratory studies have shown that at least 80% of natural isolates of *E.coli* from diverse environments worldwide can exhibit stress-induced mutagenesis [35], highlighting its significance as an essential evolutionary dynamic in the realm of microbiology. Knowing the extensive effects of stress-induced mutagenesis in pathogenic and neoplastic systems is vital for developing strategies to combat their adaptive capabilities and improve therapeutic interventions [36].

Here, due to the Schrödinger analogy, we can conveniently employ two different methods: the Rayleigh–Ritz variational method [37,38] to estimate the stationary population number, and the Rayleigh–Schrödinger perturbation theory [39] to assess the tendency of population change resulting from stress-induced mutagenesis. For an unknown system evolving under a given Hamiltonian, the Rayleigh–Ritz variational method consists of finding trial wave functions that minimize the energy of the system to approximate the unknown ground state. Hence, we can estimate the stationary population size for a family of single-peak landscapes in all dimensions through this method. On the other hand, we also study the stress-induced mutagenesis using perturbation theory on a well-known system: starting from a Hamiltonian with a well-known ground state and eigenenergy, we probe the evolution of this system perturbed by a small addition to this Hamiltonian using a power series expansion to the known ground state and the operator representation of the additional Hamiltonian perturbation. This is the Rayleigh–Schrodinger method, and we utilize it to consider two distinct extremes: gradual increases in mutation rate with stress and a sharp mutational burst when stress levels surpass a certain fitness threshold. We

demonstrate in an unchanging environment that, unlike in the former case, the latter consistently leads to a net gain in the total population size. This finding offers an explanation for the frequent appearance of mutational switches observed in nature [32,33].

## 2. Evolutionary Landscape and Ecological Influence

The landscape is typically represented as a multi-dimensional Euclidean space $\mathbb{R}^{\mathscr{D}}$, where each point $\vec{x}$ represents a unique combination of $\mathscr{D}$ scalar-strategy genetic traits [16]. When the maximum fitness $R(\vec{x})$ (which represents the selection pressure) remains constant over time [15,18], the population distribution density $b(\vec{x}, t)$ within this landscape evolves via the Fokker–Planck equation [14],

$$\partial_t b = \nabla^2(Db) + Rb ,\tag{1}$$

where the effective diffusivity $D$ represents the local speed of mutations. In other words, a higher value of $D$ results in a faster population diversification.

Our mathematical model is still incomplete as it assumes unlimited population growth. In any natural ecological system, the population growth of an organism should be limited by the resources available in its environment. The logistic model of population growth provides a better description of the population dynamics in a finite environment by taking into account the carrying capacity of the environment [7]. The carrying capacity $K$ in the logistic model of population growth [8,9] represents the maximum number of individuals that can be sustained in a given environment. When the population size approaches the carrying capacity, the growth rate decreases until the population stabilizes at the carrying capacity. This carrying capacity can be incorporated into the mathematical framework by modifying Equation (1) into an integro-differential equation,

$$\partial_t b = \nabla^2(Db) + \left[ 1 - \frac{\int d^{\mathscr{D}}\vec{x}\, b(\vec{x}, t)}{K} \right] Rb ,\tag{2}$$

in which the integration of population density distribution is the total population size $B(t) = \int d^{\mathscr{D}}\vec{x}\, b(\vec{x}, t)$. We can define the metric for population success as [40]

$$S(t) = \frac{B(t)}{K} = \frac{\int d^{\mathscr{D}}\vec{x}\, b(\vec{x}, t)}{K} ;\tag{3}$$

then, the expression for the growth rate is just $G = (1 - S)R$. This is a valid description at $S \leq 1$, and it also exhibits a decrease in not only birth but also death rate at a large population. For an example, bacteria such as *E.coli* can signal each other via quorum sensing, which can lead to a collective slowdown in metabolic rate at a dense bacterial population [41]. In general, at high cell densities, the rate of cell death may decrease due to various reasons. One factor contributing to decreased cell death is the activation of stress responses and mechanisms that enhance cell survival. Bacteria can sense and respond to stressful conditions, such as nutrient limitation or high cell density, by activating protective mechanisms that increase cell viability and reduce cell death. This adaptive response can help bacteria survive and maintain population stability in crowded environments, which has been observed with bacteria living in biofilms [42]. We rearrange the terms in Equation (2) and define a rescaled time $\tilde{t} = 2Dt$; in this way, we can arrive at:

$$-\partial_{\tilde{t}} b = \left( -\frac{1}{2}\nabla^2 - \frac{1 - S}{2D} R \right) b ,\tag{4}$$

which has the form of a hyperbolic differential equation if the success is treated as a constant.

In order to describe stress-induced mutagenesis, the effective diffusivity $D$ should not be a constant. This evolutionary regime is a major concern for medical research due to

its ability to accelerate the development of drug resistance in pathogenic and neoplastic systems [13,43,44], while also creating other complications in the treatment of infectious diseases [45]. According to the World Health Organization, antibiotic-resistant infections caused an estimated 1.27 million deaths worldwide in 2019 [46]. Stress-induced mutagenesis has also been found to have a notable impact on the evolution of the SARS-CoV-2 virus and the emergence of novel variants [47], highlighting the need for better understanding. One of the most crucial biological inquiries one could present regarding stress-induced mutagenesis is why it behaves in the way it does. There are many possible functional dependencies of mutation rate on stress, yet nature somehow seems to favor a mutational switch [32,33]. To explore this further, we cast this question into the mathematical framework presented by Equation (4). To capture the intricacies of stress-induced mutagenesis, we incorporate a heterogeneous effective diffusivity into the landscape. We seek to investigate the theoretical distinctions between the outcomes of these two different possibilities for diffusivity $D$ as a function of fitness, which is defined as the ability to reproduce. In the gradual case, a linearity governs:

$$D_{\text{gradual}}[R] = D_{\text{gradual}}^{(0)} + D_{\text{gradual}}^{(1)} \Delta R \,, \tag{5}$$

in which $\Delta R = \max(R) - R$ is the difference between the maximum fitness $\max(R)$ and the fitness $R$, whereas the sharp case follows a Heaviside step-function in which the transition happens right at the boundary between the fit and the unfit regions:

$$D_{\text{sharp}}[R] = D_{\text{sharp}}^{(0)} + D_{\text{sharp}}^{(1)} \Theta(-R) \,. \tag{6}$$

$\Theta(\zeta)$ is the Heaviside function, in which $\Theta(\zeta < 0) = 0$ and $\Theta(\zeta > 0) = 1$. Figure 1 serves as a visual representation of the basic postulations underlying our theoretical analysis. We summarize the biophysical quantities used in our mathematical model in Appendix A.

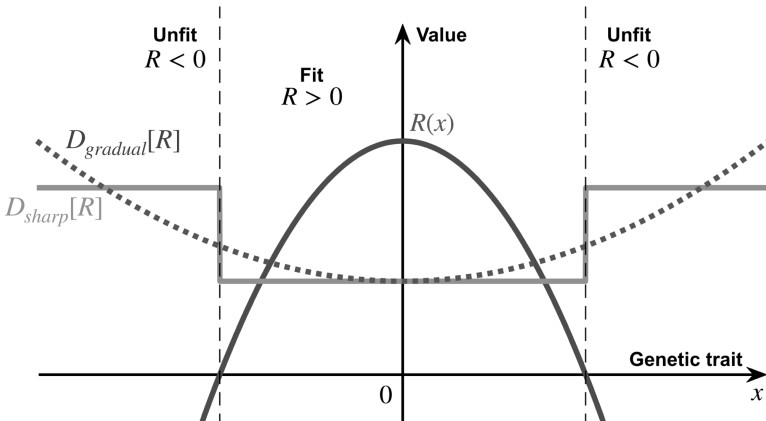

**Figure 1. The evolutionary landscape of our stress-induced mutagenesis model, illustrated around a local peak fitness.** In this work, we consider two distinct regimes of stress-induced mutagenesis, corresponding to two different heterogeneous diffusion profiles: a gradual increased diffusivity $D_{\text{gradual}}[R]$ and a sharp transition diffusivity $D_{\text{sharp}}[R]$ at the transition from the fit to the unfit phenotype. In this illustration, we used Equation (10) for $R(x)$, Equation (5) for $D_{\text{gradual}}[R]$, and Equation (6) for $D_{\text{sharp}}[R]$. Here, $\max(R) = R(0)$, and for comparison between the two cases we use $D_{\text{gradual}}^{(0)} = D_{\text{sharp}}^{(0)}$.

## 3. An Analogy to the Schrödinger Equation

The analogy between the Fokker–Planck equation as in Equation (4) and the Schrödinger equation [19,21] can be elucidated by considering the following identifications. We introduce an imaginary time variable $i\tau$ related to the diffusion coefficient $D$ and the physical

time $t$ and an energy potential $V(S, \vec{x})$ related to the population success $S$ maximum growth rate $R(\vec{x})$:

$$i\tau \leftrightarrow \tilde{t} , \ V(S, \vec{x}) \leftrightarrow -\frac{1-S}{2D} R(\vec{x}) . \tag{7}$$

This procedure of changing from real time to imaginary time is known as Wick rotation [48]. If we treat $S$ as a constant parameter, then, for Planck constant $\hbar = 1$ and mass $m = 1$, we can recast Equation (4) into a form that closely resembles the Schrödinger equation in imaginary time $\tau$:

$$i\hbar\partial_\tau \Psi = \hat{H}\Psi , \ \hat{H} = \frac{\hat{p}^2}{2m} + V(S, \vec{x}) , \tag{8}$$

where $\hat{p} = -i\hbar\nabla$ is the momentum operator and $\Psi(\vec{x}, t) \propto b(\vec{x}, t)$ represents the wavefunction of a single quantum mechanical particle of mass $m$ moving in our multi-dimensional landscape. The Hamiltonian operator $\hat{H}$ governs the behavior of this particle under the influence of the energy potential $V(S, \vec{x})$. From here on, we drop $\hbar$ and $m$ out of the analysis.

While this analogy is not exact, as it assumes that success $S$ is unchanging, independent of distribution density $b(\vec{x}, t)$, and therefore neglects the influence of the total population number on the potential energy function, it can still provide a powerful framework for understanding the dynamics of evolution. We demonstrate this by looking at the stationary state, where $b = b_{st}(\vec{x})$ is an unchanging spatial-function and thus $S = S_{st}$ is fixed. We now have an exact correspondence between Equation (8) and a time-independent Schrödinger equation associated with $E = 0$ eigenstate (also known as the stationary Schrödinger equation):

$$0 = \left( -\frac{1}{2}\nabla^2 - \frac{1-S_{st}}{2D} R \right) b_{st} \ \longleftrightarrow \ E\Psi_{st} = \left( \frac{\hat{p}^2}{2m} + V(S_{st}, \vec{x}) \right) \Psi_{st} . \tag{9}$$

Here, we obtain a powerful constraint—the stationary population success $S_{st}$ must correspond to an energy potential $V(S_{st}, \vec{x})$ that has a zero eigenenergy. Moreover, since the population density $b_{st}(\vec{x})$ is a non-negative physical field, its associated wavefunction $\Psi_{st}(\vec{x})$ should not change sign and cross zero anywhere on the entire landscape (one exception is at impenetrable boundaries, where the wave function is forced to vanish) [49]. This further restriction implies that the wavefunction should also be the ground state of the energy potential $V(S_{st}, \vec{x})$. Together, we require $V(S_{st}, \vec{x})$ to be a potential energy function that possesses a ground state with zero energy, and $\Psi(\vec{x})$ must be the ground-state wavefunction $\Psi_\Omega(\vec{x})$.

This kind of quantum mechanical analogy in classical biological phenomena has been discovered in other contexts as well. For instance, the very same Schrödinger equation we consider in our paper emerges in bacterial chemotaxis [50] as well, although only at a very special subset—but experimentally has been observed in actual bacteria populations—of the parameter space.

Let us show how to utilize this convenient constraint in practice. WE consider an inverse quadratic maximum growth rate $R(x)$ peaked and centered around the optimal combinations of genetic traits which is chosen to be at $\vec{x}_{op} = 0$:

$$R(\vec{x}) = R_0 \left[ 1 - \left( \frac{|\vec{x}|}{\lambda} \right)^2 \right] \tag{10}$$

It means the further away from the origin, the less fit an organism becomes. We call $|\vec{x}| < \lambda$ the *fit region* where $R > 0$, and $|\vec{x}| > \lambda$ the *unfit region* where $R < 0$ [40], as already shown in Figure 1. Following Equation (7), this fitness landscape corresponds to a simple harmonic oscillator potential energy $U_{SHO}(\vec{x})$ up to a shift $U_0$:

$$V(S_{st}, \vec{x}) = U_{SHO}(\vec{x}) + U_0 , \ U_{SHO}(\vec{x}) = \frac{1}{2}\omega^2|\vec{x}|^2 , \tag{11}$$

in which the angular oscillation frequency and the downward shift are

$$\omega^2 = \frac{1 - S_{st}}{D\lambda^2}R_0 \, , \; U_0 = -\frac{1}{2}\omega^2\lambda^2 \, . \tag{12}$$

The ground-state energy of a $\mathscr{D}$-dimensional oscillator, which corresponds to the purely quadratic potential $U_{SHO}(\vec{x})$, is a fundamental result that can be found in pretty much every quantum mechanics textbook [51–54]):

$$E_\Omega = \mathscr{D}\frac{\omega}{2} \, . \tag{13}$$

Therefore, for it to be zero after the energy shift, we need

$$E = E_\Omega + U_0 = 0 \implies \omega = \frac{\mathscr{D}}{\lambda^2} \, , \tag{14}$$

which directly offers us the stationary population success from Equation (12):

$$S_{st} = 1 - \frac{D\lambda^2}{R_0}\omega^2 = 1 - \frac{\mathscr{D}^2 D}{R_0\lambda^2} \, . \tag{15}$$

If $S_{st} < 0$, it means there is no sustainable success, and the population eventually becomes extinct on such ecological system. We obtain the stationary population distribution density $b_{st}(\vec{x})$, starting from the Gaussian ground-state wavefunction of a simple harmonic oscillator [51–54]:

$$b_{st}(\vec{x}) \propto \Psi_\Omega(\vec{x}) \propto \exp\left(-\frac{1}{2}\omega|\vec{x}|^2\right) = \exp\left[-\frac{\mathscr{D}}{2}\left(\frac{|\vec{x}|}{\lambda}\right)^2\right] \, . \tag{16}$$

Using Equation (3), we can determine the pre-factor and obtain

$$b_{st}(\vec{x}) = \frac{K}{\sqrt{2\pi\lambda^2/\mathscr{D}}^{\mathscr{D}}}\left(1 - \frac{\mathscr{D}^2 D}{R_0\lambda^2}\right)\exp\left[-\frac{\mathscr{D}}{2}\left(\frac{|\vec{x}|}{\lambda}\right)^2\right] \, . \tag{17}$$

Similar analytical investigations can be conducted to extract $S_{st}$ and $b_{st}(\vec{x})$ from the quantum mechanical ground-state for any function $R(\vec{x})$ defined on the landscape.

We can utilize the Rayleigh–Ritz variational method [37,38] to estimate the upper bound (and also the Weinstein method [55,56] for the lower bound) of the population success $S_{st}$ in any dimension. As a demonstration, we consider a class of landscapes with power-law dependency fitness $R(\vec{x}) = R_0[1 - (|\vec{x}|/\lambda)^\gamma]$. The fitness we considered before, given by Equation (10), belongs to this class and corresponds to the exponent value $\gamma = 2$. For a general value of $\mathscr{D}$ and $\gamma$, the exact solution for the ground state is not known. But using a Gaussian ansatz-wavefunction, we can quickly estimate the stationary population success $S_{st}$ and the width $\sigma$ of the population distribution around the optimal peak on the landscape:

$$S_{st} \geq S_{st}^{(RR)} = 1 - \frac{2D}{R_0\lambda^2}(2+\gamma)^{\frac{2+\gamma}{\gamma}}\left(\frac{\mathscr{D}}{4\gamma}\right)\left[\frac{\Gamma\left(\frac{\mathscr{D}+\gamma}{2}\right)}{2\Gamma\left(\frac{\mathscr{D}}{2}\right)}\right]^{\frac{2}{\gamma}} \, , \; \sigma \approx \left[\frac{\mathscr{D}}{2\gamma}\frac{\Gamma\left(\frac{\mathscr{D}}{2}\right)}{\Gamma\left(\frac{\mathscr{D}+\gamma}{2}\right)}\right]^{\frac{1}{2+\gamma}} \, . \tag{18}$$

We carry out in detail this estimation with a Gaussian trial wavefunction in Appendix B.1. The width $\sigma$ decreases with the exponent $\gamma$, which is consistent with our simulation findings as shown in Figure 2A. There are also other methods for estimating the ground state, such as the lesser known Weinstein method [55] and the Temple method [57,58] which can offer us lower bounds and the one-dimensional Wentzel–Kramers–Brillouin approximation [59–64] which does not admit a simple higher-dimensional generalization but has gained more

interest recently via the exact quantization condition [65–68]. We examine the Weinstein method and the Wentzel–Kramers–Brillouin approximation in Appendices B.2 and B.3. In Figure 2B, we compare estimations for $S_{st}$ using different methods with results from the simulation. We describe our simulation in Appendix C.

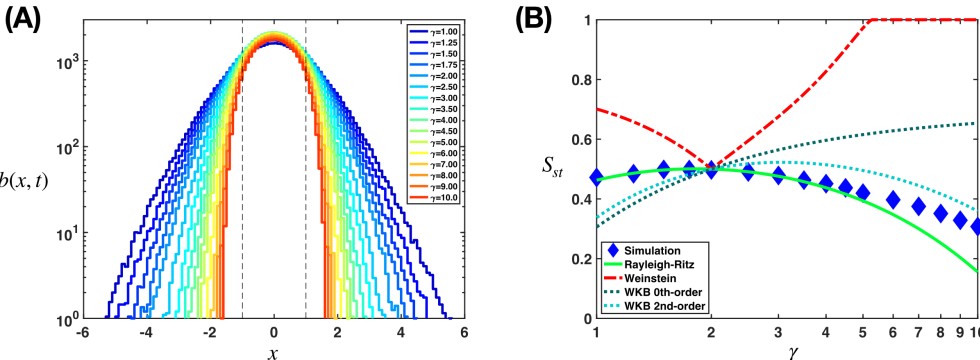

**Figure 2. Stationary population distributions and successes for different fitness landscapes can be estimated with the Rayleigh–Ritz variational method.** (**A**) The distributions of the heterogeneous population on different fitness landscapes at the stationary state, where the fitness obeys $R(\vec{x}) = R_0 \left[ 1 - \left( \frac{|\vec{x}|}{\lambda} \right)^\gamma \right]$ and the exponent $\gamma \in [1, 10]$, concentrates more around the optimal position $x_{op} = 0$ with increasing $\gamma$. The dash lines mark $x = \pm\lambda$, where the fitness hits 0. Here, we show the results from a simulation, which we described in Appendix C. Here, we consider $\mathscr{D} = 1$-dimensional landscapes and use the parameter values $D = 1/2$, $R_0 = 1$, $\lambda = 1$, and $K = 10^5$. (**B**) We compare different methods of estimation for the stationary population success with the simulation findings. We show the analytical results as obtained from the Rayleigh–Ritz variational method as in Equation (18), the Weinstein method as in Equation (A23), and the Wentzel–Krammers–Brillouin approximation (zeroth- and second-order) as in Equations (A31) and (A36).

In Figure 3, we show the analytical results obtained from the estimations of $S_{st}$, with the Rayleigh–Ritz variational method as in Equation (18), the Weinstein method as in Equation (A23), and the Wentzel–Krammers–Brillouin approximation (zeroth- and second-order) as in Equations (A31) and (A36). The Rayleigh–Ritz variation method, despite its simplicity, captures correctly and consistently the non-monotonic behavior of $S_{st}(\gamma)$ at small exponent $\gamma$, which peaks at $\gamma \approx 1.83$, and deviations from the simulation values for the region $\gamma \in [1, 4]$ are less than 4%. Biophysically, small values of $\gamma$ can be interpreted as corresponding to weakly curved fitness landscapes where fitness does not drastically decrease with mutations that move away from the optimal position. The Weinstein method, which follows naturally from the Rayleigh–Ritz variational method, requires much more calculations, and in general provides a bad estimation (except for around $\gamma \to 2$). We also explain the sudden change at $\gamma = 2$ of its estimated $S_{st}$ in Appendix B.2. The Wentzel–Krammers–Brillouin approximation is very off at the zeroth order (except also for $\gamma \to 2$), but can become much better at the second order. We note that this inability to describe the ground state at high precision is a feature expected from this approximation, which works best at highly excited states where the wavelengths are much smaller than that of the potential characteristic length scale [69].

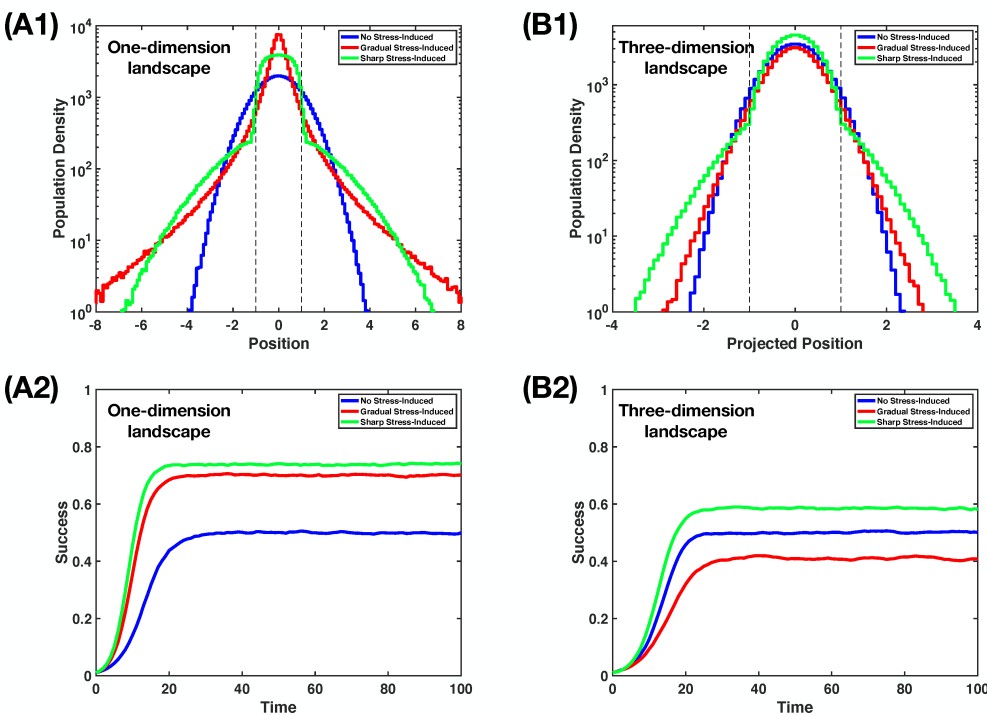

**Figure 3. Rayleigh–Schrodinger perturbation theory can predict how different regimes of stress-induced mutagenesis affect the population success, extrapolatable beyond the perturbative regime.** Here, we show our simulation findings, in which we use parameter values $R_0 = 1$, $\lambda = 1$, $K = 10^5$, and fitness as in Equation (10). We describe our simulation in Appendix C. (**A1**) The population distributions at a stationary state on $\mathscr{D} = 1$-dimensional landscape for no stress-induced, gradual stress-induced, and sharp stress-induced mutagenesis regimes. The dash lines mark $x = \pm\lambda$. For a gradual stress-induced regime, we use Equation (21) with $\epsilon = 10$; for a sharp stress-induced regime, we use Equation (28) with $\epsilon = 10$. (**A2**) The evolution of population success $S(t)$ with time $t$ on $\mathscr{D} = 1$-dimensional landscape for different mutagenesis regimes. (**B1**) The population distributions at stationary state on $\mathscr{D} = 3$-dimensional landscape. We consider looking at the distribution from a projection, i.e., $\vec{x} = (x_1, x_2, x_3)$; then, we can use $x_1$ as the projected position. The dash lines mark $x_1 = \pm\lambda$. For gradual stress-induced regime, we use Equation (21) with $\epsilon = 0.1$. Note that with $\epsilon = 10$, the population becomes extinct, as $S_{\text{st}} = 0$. For a sharp stress-induced regime, we use Equation (28) with $\epsilon = 10$. (**B2**) The evolution of population success $S(t)$ with time $t$ on $\mathscr{D} = 3$-dimensional landscape.

## 4. Applying the Rayleigh–Schrödinger Perturbation Theory to Stress-Induced Mutagenesis

We investigate the impact of stress-induced mutagenesis on the stationary population size $B_{\text{st}}$ in both cases, as listed in Equations (5) and (6), for all natural dimensionality $\mathscr{D} \in \mathbb{N}$ of the landscape. Rather than attempting to solve the exact, non-tractable evolution dynamics on the landscape, we adopt a perturbative approach. This enables us to reveal distinctions between the two cases in a tractable manner. We split the Hamiltonian in Equation (8) into the unperturbed $\hat{H}_0$ and the perturbed $\epsilon\hat{H}_p$. At the stationary state,

$$\hat{H} = \hat{H}_0 + \epsilon\hat{H}_p \,, \ \hat{H}_0 = \frac{1}{2}\hat{p}^2 + V(S_{\text{st}}, \vec{x}) \,, \tag{19}$$

where the energy potential $V(S_{\text{st}}, \vec{x})$ is as given in Equations (11) and (12). At the lowest order of perturbation $\mathscr{O}(\epsilon)$, the correction $\epsilon\delta E_\Omega$ to the ground-state energy in Equation (13)

can be estimated via the Rayleigh–Schrödinger perturbation theory even for a non-Hermittian $\hat{H}_p$ [39]:

$$E_\Omega = \mathscr{D}\frac{\omega}{2} + \epsilon \delta E_\Omega^{(1)} \;,\; \delta E_\Omega^{(1)} = \frac{\int d^{\mathscr{D}}\vec{x}\, \Psi_\Omega^\dagger(\vec{x}).\hat{H}_p.\Psi_\Omega(\vec{x})}{\int d^{\mathscr{D}}\vec{x}\, \Psi_\Omega^\dagger(\vec{x})\Psi_\Omega(\vec{x})} \;, \tag{20}$$

where $\Psi_\Omega(\vec{x})$ is the ground-state wavefunction of the unperturbed Hamiltonian as given by Equation (16). Our analysis can unveil the tendencies with which different manifestations of stress-induced mutagenesis affect the population, either boosting or suppressing success $S_{\text{st}}$.

### 4.1. A Gradual Change

The diffusivity on the landscape as in Equation (5), which is associated with a gradual change in mutation rates, can be expressed as follows:

$$D_{\text{gradual}} = D\left[1 + \epsilon\left(\frac{|\vec{x}|}{\lambda}\right)^2\right] \;, \tag{21}$$

where we define constants $D$ and $\epsilon$ to be

$$D = D_{\text{gradual}}^{(0)} \;,\; \epsilon = \frac{D_{\text{gradual}}^{(1)} R_0}{D_{\text{gradual}}^{(0)}} \;. \tag{22}$$

We consider $\epsilon$ as a perturbation parameter in limit $\epsilon \ll 1$ which corresponds to $D_{\text{gradual}}^{(0)} \gg D_{\text{gradual}}^{(1)} R_0$.

The perturbed Hamiltonian in Equation (19) for this regime of stress-induced mutagenesis is given by a non-Hermitian operator:

$$\hat{H}_p = \frac{1}{2}\hat{p}^2\left(\frac{|\vec{x}|}{\lambda}\right)^2 \;. \tag{23}$$

Applying Equation (20), we obtain

$$\delta E_\Omega^{(1)} = \frac{1}{8\lambda^2}\mathscr{D}(\mathscr{D}-2) \;. \tag{24}$$

The detail of this calculation can be found in Appendix D.1.

Applying Equation (14) including the ground-state energy correction,

$$E = E_\Omega + U_0 = \mathscr{D}\frac{\omega}{2} + \epsilon\frac{1}{8\lambda^2}\mathscr{D}(\mathscr{D}-2) - \frac{1}{2}\omega^2\lambda^2 = 0 \;, \tag{25}$$

we can approximate $\omega$ at the first order of $\epsilon$-expansion:

$$\omega \approx \frac{\mathscr{D}}{\lambda^2}[1 + \epsilon\eta(\mathscr{D})] \;,\; \eta(\mathscr{D}) = \frac{\mathscr{D}-2}{4\mathscr{D}} \;. \tag{26}$$

For a high dimensionality $\mathscr{D} > 2$, $\eta(\mathscr{D})$ is a positive value. Perturbative stress-induced mutagenesis in this regime increases $\omega \uparrow$. Since Equation (12) indicates that $\omega$ and $S_{\text{st}}$ have an inverse monotonic relationship, this means we obtain a reduction in success $S_{\text{st}} \downarrow$. In other words, stress-induced mutagenesis tends to suppress the population success when the number of relevant genetic traits on the landscape is high.

The above statement does not change if we consider another power-law dependency for the perturbative Hamiltonian in Equation (23), since

$$\hat{H}_p \propto \hat{p}^2|\vec{x}|^\kappa \implies \eta(\mathscr{D}) \propto \mathscr{D} - \kappa \;. \tag{27}$$

We obtain $\eta(\mathscr{D}) > 0$ when $\mathscr{D} > \kappa$. The derivation can be found in Appendix D.2.

*4.2. A Sharp Change*

We can rewrite Equation (6), which describes the diffusivity on the landscape associated with a sharp change in mutation rates, as follows:

$$D_{\text{sharp}} = D\left[1 + \epsilon\Theta\left(\frac{|\vec{x}|}{\lambda} - 1\right)\right] . \tag{28}$$

From Equations (6) and (10), we define the constants $D$ and $\epsilon$ to be

$$D = D_{\text{sharp}}^{(0)} , \quad \epsilon = \frac{D_{\text{sharp}}^{(1)}}{D_{\text{sharp}}^{(0)}} . \tag{29}$$

Here, we treat the up-step contribution as perturbation $\epsilon \ll 1$, which requires $D_{\text{sharp}}^{(0)} \gg D_{\text{sharp}}^{(1)}$ for subsequent calculations to be analytically tractable.

For this posibility of stress-induced mutagenesis, the perturbed Hamiltonian in Equation (19) is given by the following non-Hermitian operator:

$$\hat{H}_p = \frac{1}{2}\hat{p}^2\Theta\left(\frac{|\vec{x}|}{\lambda} - 1\right) . \tag{30}$$

Following Equation (20), we can make the following estimation:

$$\delta E_\Omega^{(1)} = \frac{1}{\lambda^2}\frac{(\omega\lambda^2)^{\frac{\mathscr{D}}{2}+1}\left[-2e^{-\omega\lambda^2} + \omega\lambda^2 \, \mathrm{E}_{-\frac{\mathscr{D}}{2}}\left(\omega\lambda^2\right)\right]}{2\Gamma\left(\frac{\mathscr{D}}{2}\right)} , \tag{31}$$

where we remind that $\omega$ is as given in Equation (12). We carry out the details of this calculation in Appendix D.3. We can then determine $\omega$ using Equation (14):

$$\begin{aligned}
E &= E_\Omega + U_0 \\
&= \mathscr{D}\frac{\omega}{2} + \epsilon\frac{1}{\lambda^2}\frac{(\omega\lambda^2)^{\frac{\mathscr{D}}{2}+1}\left[-2e^{-\omega\lambda^2} + \omega\lambda^2 \, \mathrm{E}_{-\frac{\mathscr{D}}{2}}\left(\omega\lambda^2\right)\right]}{2\Gamma\left(\frac{\mathscr{D}}{2}\right)} - \frac{1}{2}\omega^2\lambda^2 = 0 ,
\end{aligned} \tag{32}$$

in which we obtain the approximate solution:

$$\omega \approx \frac{\mathscr{D}}{\lambda^2}[1 + \epsilon\eta(\mathscr{D})] , \ \eta(\mathscr{D}) = \frac{\mathscr{D}^{\frac{\mathscr{D}}{2}-1}\left[-2e^{-\mathscr{D}} + \mathscr{D} \, \mathrm{E}_{-\frac{\mathscr{D}}{2}}(\mathscr{D})\right]}{\Gamma\left(\frac{\mathscr{D}}{2}\right)} , \tag{33}$$

where $\mathrm{E}_{...}(\ldots)$ is the generalized exponential integral [70]. Since $\eta(\mathscr{D}) < 0$ for every natural dimensionality $\mathscr{D} \in \mathbb{N}$, this perturbative stress-induced mutagenesis effect always reduces $\omega \downarrow$, thus increasing success $S_{\text{st}} \uparrow$ as follows from Equation (12).

The nonperturbative stationary solution of this case has been studied in our previous work [71]. For a sanity check, one can show that what we found here using the Rayleigh–Schrödinger perturbation theory agrees with the exact result at the leading order of perturbative parameters $\epsilon$.

*4.3. A Comparison between Two Stress-Induced Mutagenesis Regimes*

While the calculations in the previous section are performed with perturbation theory, which corresponds to weak stress-induced mutagenesis effects, our findings move beyond that, as shown with simulations in Figure 3 for large values of $\epsilon$. We describe our simulation

in Appendix C. Gradual stress-induced mutagenesis, which can be beneficial on low-dimensional landscapes (see Figure 3A1,A2), becomes quite lethal at high-dimensional landscapes (see Figure 3B1,B2). Sharp stress-induced mutagenesis, on the other hand, always up the stationary population success.

To gain some intuitive understanding of how these two regimes can be so different, consider the following. The expression for the total diffusive flux of population on the landscape is given by $J = \nabla(Db)$, which can be further broken down into two contributions: the diffusion gradient contribution $J_{\text{diff}} = (\nabla D)b$ which is driven by the local slope of the diffusivity, and the density gradient contribution $J_{\text{dens}} = D(\nabla b)$ which is generated by the heterogeneity of population distribution. In the case of a gradually changing $D_{\text{gradual}}[R]$, contribution $J_{\text{diff}}$ exists generally everywhere, pointing towards the optimal combination of genetic traits. This means that there is a clear guidance toward peak fitness on the $\mathscr{D}$-dimensional space of genetic variations, focusing the population into a specific hypervolume. In contrast, for a sharp transition $D_{\text{sharp}}[R]$, this flux vanishes everywhere except at the $(\mathscr{D} - 1)$-dimensional boundary hypersurface between fit and unfit regions. Thus, intuitively, we expect that abruptly increasing mutation rates via stress-induced mutagenesis may be less effective in maintaining a large stationary population size.

Our application of Rayleigh–Schrödinger perturbation theory quickly revealed a paradoxical outcome that influences evolutionary dynamics in the presence of multiple relevant evolving genetic traits. We interpret this mathematical finding as follows: in the case of gradual stress-induced mutagenesis, the diffusive gradient flux becomes less effective at population concentration towards the fit region, as it spreads out excessively as the number of landscape dimensions increases. Conversely, sharp stress-induced mutagenesis enables the diffusive gradient flux to remain spatially focused, even singular, and thus boosts the population success consistently, regardless of the number of genetic traits involved. It has been observed that natural selection favors species that can optimize multiple biological capabilities simultaneously [5,72]. As a result, the sharp regime of stress-induced mutagenesis may be preferred. Empirical evidence supports this notion [32,33]. Here, we showed a quantitative argument for why this might be the case.

## 5. Discussion

In this study, we reveal a curious analogy between a fundamental equation in quantum mechanics and the equation that governs the evolution of multiple genetic traits in a population. We show that determining the stationary distribution of a heterogeneous population can be mapped to the problem of finding the ground state wavefunction, fostering a more unified understanding of diverse phenomena and leading to new analytical approaches. Techniques developed for dealing with quantum mechanical systems can be adapted and applied to comprehend population dynamics, not only more expeditiously but also more profoundly.

The Rayleight–Ritz variational method [37,38] allows us quick estimation of the population number at an equilibrium state, and also approximation of the genotypic diversity in the population, usually with a test function, such as a Gaussian shape, where the most dominant genotype (the mean) and the heterogeneity (the width) are well-defined. In standard coarse-graining macroscopic description of evolutionary game theory with competing species, such as in the study of cancer progression [73,74] and optimizing chemotherapeutic treatment [75] via the G-function [16,76], these two features (the mean and the width) are the most important mesoscopic variables.

Perhaps even more interesting, the Rayleigh–Schrödinger perturbation theory [39] can be utilized to answer a biological "why" question. Specifically, we showed quantitative supporting evidence for why stress-induced mutagenesis exhibits a sharp transition rather than a gradual change in mutation rates, which is commonly observed in microbial and cancer cells [32,33]. In contrast to physics, where phenomena may arise spontaneously, the complex emergent behavior observed in biology is the product of billions of years of natural selection, which has relentlessly honed and optimized the living systems we see

today [1,2]. It is therefore essential to focus on understanding why biological phenomena occur, rather than simply how they occur [77,78]. Our findings highlight the significance of our approach as a valuable framework for modeling biological evolution, rather than just a mere mathematical exercise.

The methodology presented in this paper opens up many avenues for future research. Although our study focused on a static ecological system, it is essential to acknowledge that most ecological systems in the real world are highly complex [79,80] and dynamic in nature [81]. Therefore, one possible direction for future research is to extend our analogy to incorporate the effects of dynamical ecological systems, such as seasonal change or periodic cycle of drug administration [40], in which it is expected that quantum mechanical methods to deal with temporal varying Hamiltonian (e.g., time-dependent perturbation theory [82], adiabatic invariant [83], Floquet theory [84]) can be employed. Another exciting adventure is to explore the impact of landscape topology. In particular, it would be interesting to investigate whether certain topological features of the fitness landscape can amplify or suppress the effects of stress-induced mutagenesis on population dynamics, since spatial topology has been shown to affect the quantization conditions greatly [85]. Finally, there have been recent attempts to investigate exotic collective behaviors and new sectors of evolutionary dynamics using robots with engineered ecological interactions [86,87] and stress-induced mutable genomes [88], which might allow for observing the realization of our analogy in a physical evolvable system beyond biology. So much to do; the future seems bright and exciting.

**Author Contributions:** Conceptualization, T.V.P. and T.K.D.; analytical investigation, D.V.T., V.D.A. and K.T.P.; writing—original draft preparation, D.V.T., V.D.A., K.T.P., T.K.D., V.H.D. and T.V.P.; writing—review and editing, D.V.T., V.D.A., K.T.P., D.M.N., T.K.D., V.H.D. and T.V.P.; visualization, T.V.P.; supervision, H.D.T., V.H.D. and T.V.P.; project administration, T.V.P. All authors provided critical feedback and helped shape the research, analysis and manuscript. All authors have read and agreed to the published version of the manuscript.

**Funding:** This research received no external funding.

**Institutional Review Board Statement:** Not applicable.

**Informed Consent Statement:** Not applicable.

**Data Availability Statement:** The MatLab codes for the simulations used in this study are available from the corresponding author upon request.

**Acknowledgments:** We thank Robert H. Austin, Kenneth J. Pienta, Joel Brown, Emma U. Hammarlund and Sarah R. Amend for the chance to give a talk on this simple but curious finding at Moffit Cancer Center (2021) and many useful discussions followed, which motivated us to share it with a wider audience. We also thank Truong H. Cai, Ramzi Khuri, Thierry Emonet, Henry Mattingly and Lam Vo for insightful comments.

**Conflicts of Interest:** The authors declare no conflict of interest.

## Appendix A. Summary of All Mathematical Quantities

Here, we summarize all quantities in our proposed model for the evolution dynamics with stress-induced mutagenesis:

- $t$: time.
- $\vec{x}$: position (a genomic configuration) in the abstract $\mathscr{D}$-dimensional fitness landscape.
- $b(\vec{x}, t)$: population density on the landscape, which has the unit of population number per unit volume (equal to a unit length to the power $\mathscr{D}$).
- $D(\vec{x})$: effective diffusivity in the landscape, which has the unit of unit length squared (to the power 2) per unit time.
- $R(\vec{x})$: the maximum growth rate of the sub-population located at position $\vec{x}$ in the landscape, which has the unit of inverse unit time.
- $K$: carrying capacity, which has the unit of population number.

- $S$: success, which is the ratio between the total population number $\int d^{\mathscr{D}} \vec{x} b(\vec{x}, t)$ and the carrying capacity $K$; therefore, it is a dimensionless quantity.

**Appendix B. Estimations of Stationary Population Success**

Let us define the following:

$$\Phi = \frac{1 - S_{\text{st}}}{2D} R_0 \lambda^{-\gamma} , \tag{A1}$$

so that the potential as in Equation (7) can be rewritten as

$$V(S_{\text{st}}, \vec{x}) = -\Phi \lambda^{\gamma} + \Phi |\vec{x}|^{\gamma} . \tag{A2}$$

We want to estimate the ground state energy $E_\Omega$ of a pure power-law potential $U(\vec{x}) = \Phi |\vec{x}|^{\gamma}$, which can be related to the ground state energy $\tilde{E}_\Omega$ of the potential $\tilde{U}(\vec{x}) = |\vec{x}|^{\gamma}$:

$$\tilde{E}_\Omega = \Phi^{\frac{2}{2+\gamma}} \tilde{E}_\Omega . \tag{A3}$$

If we can estimate $\tilde{E}_\Omega$, then we can estimate the stationary population success $S_{\text{st}}$ via the equality Equation (14):

$$-\Phi \lambda^{\gamma} + E_\Omega = 0 \implies S_{\text{st}} = 1 - \frac{2D}{R_0 \lambda^2} \tilde{E}_\Omega^{\frac{2+\gamma}{\gamma}} . \tag{A4}$$

We note that it is possible that the mathematical estimation of $S_{\text{st}}$ can become smaller than zero or larger than one, which is physically impossible. In that case, we can interpret these results as $S_{\text{st}}^{(\text{estimation})} \to 0$ if $S_{\text{st}}^{(\text{estimation})} < 0$ (population extinction), or $S_{\text{st}}^{(\text{estimation})} \to 1$ if $S_{\text{st}}^{(\text{estimation})} > 1$.

For simplicity, we work with $\tilde{U}(\vec{x})$ instead of $U(\vec{x})$. The relationship between the ground states $\Psi_\Omega(\vec{x})$ and $\tilde{\Psi}_\Omega(\vec{x})$ are given by

$$\Psi_\Omega(\vec{x}) = \Phi^{\frac{\mathscr{D}}{2+\gamma}} \tilde{\Psi}_\Omega \left( \Phi^{-\frac{1}{2+\gamma}} \vec{x} \right) . \tag{A5}$$

*Appendix B.1. Application of the Rayleigh–Ritz Variational Method*

For the Rayleigh–Ritz variational method [37,38], we need a trial wavefunction. We choose a Gaussian function centered at $\vec{x} = 0$ and that has the standard deviation $\sigma$ as a parameter:

$$\tilde{\Psi}_{\text{trial}}(\sigma, \vec{x}) \propto \exp \left( -\frac{|\vec{x}|^2}{2\sigma^2} \right) . \tag{A6}$$

An upper bound for $\tilde{E}_\Omega$ can be estimated via the following minimization with respect to $\sigma$:

$$\begin{aligned} \tilde{E}_\Omega &\leq \min_\sigma \left[ \frac{\int d^{\mathscr{D}} \vec{x} \tilde{\Psi}_{\text{trial}}(\sigma, \vec{x}) \hat{H} \tilde{\Psi}_{\text{trial}}(\sigma, \vec{x})}{\int d^{\mathscr{D}} \vec{x} \tilde{\Psi}_{\text{trial}}(\sigma, \vec{x}) \tilde{\Psi}_{\text{trial}}(\sigma, \vec{x})} \right] \\ &= \min_\sigma \left\{ \frac{\int d^{\mathscr{D}} \vec{x} \tilde{\Psi}_{\text{trial}}(\sigma, \vec{x}) \left[ -\frac{1}{2} \nabla^2 + \tilde{U}(\vec{x}) \right] \tilde{\Psi}_{\text{trial}}(\sigma, \vec{x})}{\int d^{\mathscr{D}} \vec{x} \tilde{\Psi}_{\text{trial}}^2(\sigma, \vec{x})} \right\} = \tilde{E}_\Omega^{(\text{RR})} . \end{aligned} \tag{A7}$$

Let us evaluate the function inside $\{...\}$:

$$F(\sigma) = \frac{\int_0^\infty d|\vec{x}||\vec{x}|^{\mathscr{D}-1} \exp\left(-\frac{|\vec{x}|^2}{2\sigma^2}\right)\left[-\frac{1}{2}\nabla^2 + |\vec{x}|^\gamma\right]\exp\left(-\frac{|\vec{x}|^2}{2\sigma^2}\right)}{\int_0^\infty d|\vec{x}||\vec{x}|^{\mathscr{D}-1}\exp\left(-\frac{|\vec{x}|^2}{\sigma^2}\right)}$$
$$= \frac{f_{\text{kin}}(\sigma) + f_{\text{pot}}(\sigma)}{\frac{1}{2}\Gamma\left(\frac{\mathscr{D}}{2}\right)\sigma^{\mathscr{D}}}. \tag{A8}$$

This has two components, the potential part,

$$f_{\text{pot}}(\sigma) = \int_0^\infty d|\vec{x}||\vec{x}|^{\mathscr{D}-1}\exp\left(-\frac{|\vec{x}|^2}{2\sigma^2}\right)|\vec{x}|^\gamma \exp\left(-\frac{|\vec{x}|^2}{2\sigma^2}\right)$$
$$= \int_0^\infty d|\vec{x}||\vec{x}|^{\mathscr{D}+\gamma-1}\exp\left(-\frac{|\vec{x}|^2}{\sigma^2}\right) = \frac{1}{2}\Gamma\left(\frac{\mathscr{D}+\gamma}{2}\right)\sigma^{\mathscr{D}+\gamma}, \tag{A9}$$

and the kinetic part,

$$f_{\text{kin}}(\sigma) = -\frac{1}{2}\int_0^\infty d|\vec{x}||\vec{x}|^{\mathscr{D}-1}\exp\left(-\frac{|\vec{x}|^2}{2\sigma^2}\right)\nabla^2 \exp\left(-\frac{|\vec{x}|^2}{2\sigma^2}\right)$$
$$= -\frac{1}{2}\int_0^\infty d|\vec{x}||\vec{x}|^{\mathscr{D}-1}\exp\left(-\frac{|\vec{x}|^2}{\sigma^2}\right)\left(\frac{|\vec{x}|^2 - \mathscr{D}\sigma^2}{\sigma^4}\right) = \frac{1}{4}\Gamma\left(\frac{\mathscr{D}+2}{2}\right)\sigma^{\mathscr{D}-2}, \tag{A10}$$

where operating the Laplacian on any function $A(|\vec{x}|)$ offers

$$\nabla^2 A(|\vec{x}|) = \frac{1}{|\vec{x}|^{\mathscr{D}-1}}\partial_{|\vec{x}|}\left[|\vec{x}|^{\mathscr{D}-1}\partial_{|\vec{x}|}A(|\vec{x}|)\right]. \tag{A11}$$

Together, with these two contributions, we have

$$F(\sigma) = \frac{\frac{1}{4}\Gamma\left(\frac{\mathscr{D}+2}{2}\right)\sigma^{\mathscr{D}-2} + \frac{1}{2}\Gamma\left(\frac{\mathscr{D}+\gamma}{2}\right)\sigma^{\mathscr{D}+\gamma}}{\frac{1}{2}\Gamma\left(\frac{\mathscr{D}}{2}\right)\sigma^{\mathscr{D}}} = \frac{\mathscr{D}}{4}\sigma^{-2} + \frac{\Gamma\left(\frac{\mathscr{D}+\gamma}{2}\right)}{\Gamma\left(\frac{\mathscr{D}}{2}\right)}\sigma^\gamma. \tag{A12}$$

The minimum of $F(\sigma)$ can be found by using the Cauchy inequality:

$$F(\sigma) = \gamma\left(\frac{\mathscr{D}}{4\gamma}\sigma^{-2}\right) + 2\left[\frac{\Gamma\left(\frac{\mathscr{D}+\gamma}{2}\right)}{2\Gamma\left(\frac{\mathscr{D}}{2}\right)}\sigma^\gamma\right]$$
$$\geq (2+\gamma)\left\{\left(\frac{\mathscr{D}}{4\gamma}\right)^\gamma\left[\frac{\Gamma\left(\frac{\mathscr{D}+\gamma}{2}\right)}{2\Gamma\left(\frac{\mathscr{D}}{2}\right)}\right]^2\right\}^{\frac{1}{2+\gamma}} = F(\sigma_{\text{min}}), \tag{A13}$$

and the equal sign appears when

$$\frac{\mathscr{D}}{4\gamma}\sigma_{\text{min}}^{-2} = \frac{\Gamma\left(\frac{\mathscr{D}+\gamma}{2}\right)}{2\Gamma\left(\frac{\mathscr{D}}{2}\right)}\sigma_{\text{min}}^\gamma \implies \sigma_{\text{min}} = \left[\frac{\mathscr{D}}{2\gamma}\frac{\Gamma\left(\frac{\mathscr{D}}{2}\right)}{\Gamma\left(\frac{\mathscr{D}+\gamma}{2}\right)}\right]^{\frac{1}{2+\gamma}}. \tag{A14}$$

Hence, we obtain an upper estimation for $\tilde{E}_\Omega$, i.e., $\tilde{E}_\Omega \leq F(\sigma_{\min}) = \tilde{E}_\Omega^{(RR)}$. We plug this inside Equation (A4), and thus we can have a lower-bound estimate of the stationary population success:

$$
\begin{aligned}
S_{\mathrm{st}} \geq S_{\mathrm{st}}^{(RR)} &= 1 - \frac{2D}{R_0 \lambda^2} \left[ \tilde{E}_\Omega^{(RR)} \right]^{\frac{2+\gamma}{\gamma}} \\
&= 1 - \frac{2D}{R_0 \lambda^2} (2+\gamma)^{\frac{2+\gamma}{\gamma}} \left( \frac{\mathscr{D}}{4\gamma} \right) \left[ \frac{\Gamma\left(\frac{\mathscr{D}+\gamma}{2}\right)}{2\Gamma\left(\frac{\mathscr{D}}{2}\right)} \right]^{\frac{2}{\gamma}} .
\end{aligned}
\tag{A15}
$$

*Appendix B.2. Application of the Weinstein Method*

The Weinstein method [55,58] picks up where the Rayleigh–Ritz has left off, using the trial wavefunction $\tilde{\Psi}(\sigma_{\min}, \vec{x})$ to estimate the lower bound of $\tilde{E}_\Omega$:

$$
\tilde{E}_\Omega \geq \tilde{E}_\Omega^{(RR)} - \left\{ \langle \hat{H}^2 \rangle_{\sigma_{\min}} - \left[ \tilde{E}_\Omega^{(RR)} \right]^2 \right\}^{\frac{1}{2}} = \tilde{E}_\Omega^{(W)} ,
\tag{A16}
$$

where $\sigma_{\min}$ is as found in Equation (A14) and

$$
\begin{aligned}
\langle \hat{H}^2 \rangle_\sigma &= \frac{\int d^{\mathscr{D}} \vec{x} \, \tilde{\Psi}_{\mathrm{trial}}(\sigma_{\min}, \vec{x}) \hat{H}^2 \tilde{\Psi}_{\mathrm{trial}}(\sigma, \vec{x})}{\int d^{\mathscr{D}} \vec{x} \, \tilde{\Psi}_{\mathrm{trial}}^2(\sigma, \vec{x})} \Bigg|_{\sigma = \sigma_{\min}} \\
&= \frac{\int_0^\infty d|\vec{x}| |\vec{x}|^{\mathscr{D}-1} \exp\left(-\frac{|\vec{x}|^2}{2\sigma^2}\right) \left[ -\frac{1}{2}\nabla^2 + |\vec{x}|^\gamma \right]^2 \exp\left(-\frac{|\vec{x}|^2}{2\sigma^2}\right)}{\frac{1}{2}\Gamma\left(\frac{\mathscr{D}}{2}\right)\sigma^{\mathscr{D}}} \Bigg|_{\sigma = \sigma_{\min}} .
\end{aligned}
\tag{A17}
$$

The numerator of this expression can be divided into four parts, evaluated separately. Opening up the operator $\left[ -\frac{1}{2}\nabla^2 + |\vec{x}|^\gamma \right]^2$, we obtain the contribution from the $|\vec{x}|^\gamma |\vec{x}|^\gamma$ term:

$$
\begin{aligned}
N_1(\sigma) &= \int_0^\infty d|\vec{x}| |\vec{x}|^{\mathscr{D}-1} \exp\left(-\frac{|\vec{x}|^2}{2\sigma^2}\right) |\vec{x}|^{2\gamma} \exp\left(-\frac{|\vec{x}|^2}{2\sigma^2}\right) \\
&= \int_0^\infty d|\vec{x}| |\vec{x}|^{\mathscr{D}+2\gamma-1} \exp\left(-\frac{|\vec{x}|^2}{\sigma^2}\right) = \frac{1}{2}\Gamma\left(\frac{\mathscr{D}+2\gamma}{2}\right)\sigma^{\mathscr{D}+2\gamma} ,
\end{aligned}
\tag{A18}
$$

the $-\frac{1}{2}|\vec{x}|^\gamma \nabla^2$ term:

$$
\begin{aligned}
N_2(\sigma) &= -\frac{1}{2} \int_0^\infty d|\vec{x}| |\vec{x}|^{\mathscr{D}-1} \exp\left(-\frac{|\vec{x}|^2}{2\sigma^2}\right) |\vec{x}|^\gamma \nabla^2 \exp\left(-\frac{|\vec{x}|^2}{2\sigma^2}\right) \\
&= -\frac{1}{2} \int_0^\infty d|\vec{x}| |\vec{x}|^{\mathscr{D}+\gamma-1} \exp\left(-\frac{|\vec{x}|^2}{\sigma^2}\right) \left( \frac{|\vec{x}|^2 - \mathscr{D}\sigma^2}{\sigma^4} \right) \\
&= \frac{1}{8}(\mathscr{D}-\gamma)\Gamma\left(\frac{\mathscr{D}+\gamma}{2}\right)\sigma^{\mathscr{D}+\gamma-2} ,
\end{aligned}
\tag{A19}
$$

the $-\frac{1}{2}\nabla^2|\vec{x}|^\gamma$ term:

$$
\begin{aligned}
N_3(\sigma) &= -\frac{1}{2}\int_0^\infty d|\vec{x}||\vec{x}|^{\mathscr{D}-1}\exp\left(-\frac{|\vec{x}|^2}{2\sigma^2}\right)\nabla^2\left[|\vec{x}|^\gamma\exp\left(-\frac{|\vec{x}|^2}{2\sigma^2}\right)\right] \\
&= -\frac{1}{2}\int_0^\infty d|\vec{x}||\vec{x}|^{\mathscr{D}+\gamma-3}\exp\left(-\frac{|\vec{x}|^2}{\sigma^2}\right) \\
&\quad \left[\frac{|\vec{x}|^4-(\mathscr{D}+2\gamma)\sigma^2|\vec{x}|^2+\gamma(\mathscr{D}+\gamma-2)\sigma^4}{\sigma^4}\right] \\
&= \frac{1}{8}(\mathscr{D}-\gamma)\Gamma\left(\frac{\mathscr{D}+\gamma}{2}\right)\sigma^{\mathscr{D}+\gamma-2},
\end{aligned}
\tag{A20}
$$

and the $\frac{1}{4}\nabla^2\nabla^2$ term:

$$
\begin{aligned}
N_4(\sigma) &= \frac{1}{4}\int_0^\infty d|\vec{x}||\vec{x}|^{\mathscr{D}-1}\exp\left(-\frac{|\vec{x}|^2}{2\sigma^2}\right)\nabla^2\left[\nabla^2\exp\left(-\frac{|\vec{x}|^2}{2\sigma^2}\right)\right] \\
&= \frac{1}{4}\int_0^\infty d|\vec{x}||\vec{x}|^{\mathscr{D}-1}\exp\left(-\frac{|\vec{x}|^2}{\sigma^2}\right)\left[\frac{|\vec{x}|^4-2(\mathscr{D}+2)\sigma^2|\vec{x}|^2+\mathscr{D}(\mathscr{D}+2)\sigma^4}{\sigma^8}\right] \\
&= \frac{1}{32}\mathscr{D}(\mathscr{D}+2)\Gamma\left(\frac{\mathscr{D}}{2}\right)\sigma^{\mathscr{D}-4}.
\end{aligned}
\tag{A21}
$$

Thus, following from Equation (A16), we obtain

$$
\tilde{E}_\Omega^{(W)}=\tilde{E}_\Omega^{(RR)}-\left\{\left.\frac{\sum_{j=1}^4 N_j(\sigma)}{\frac{1}{2}\Gamma\left(\frac{\mathscr{D}}{2}\right)\sigma^{\mathscr{D}}}\right|_{\sigma=\sigma_{\min}}-\left[\tilde{E}_\Omega^{(RR)}\right]^2\right\}^{\frac{1}{2}},
\tag{A22}
$$

in which we can estimate the upper bound for the stationary population success as

$$
S_{\text{st}}\leq S_{\text{st}}^{(W)}=1-\frac{2D}{R_0\lambda^2}\left[\tilde{E}_\Omega^{(W)}\right]^{\frac{2+\gamma}{\gamma}}.
\tag{A23}
$$

In Figure 2, for the dependence of $S_{\text{st}}^{(W}$ on the exponent $\gamma$, we see a sharp turn at $\gamma=2$, where $\tilde{E}^{(W)}=\tilde{E}^{(RR)}$. To understand this, let us take a look at how the difference between them progresses with $\gamma$ by rewriting Equation (A16) as follows:

$$
\tilde{E}_{\text{st}}^{(RR)}-\tilde{E}_{\text{st}}^{(W)}=\left[\langle\hat{H}^2\rangle_\gamma-\langle\hat{H}\rangle_\gamma^2\right]^{\frac{1}{2}},
\tag{A24}
$$

in which we use

$$
\begin{aligned}
\langle\hat{H}^2\rangle_\gamma &= \left\langle\Psi_{\text{trial}}[\sigma_{\min}(\gamma)]\middle|\hat{H}^2\middle|\Psi_{\text{trial}}[\sigma_{\min}(\gamma)]\right\rangle, \\
\langle\hat{H}\rangle_\gamma^2 &= \left\langle\Psi_{\text{trial}}[\sigma_{\min}(\gamma)]\middle|\hat{H}\middle|\Psi_{\text{trial}}[\sigma_{\min}(\gamma)]\right\rangle^2.
\end{aligned}
\tag{A25}
$$

Since $\langle\hat{H}^2\rangle_\gamma$ and $\langle\hat{H}\rangle_\gamma^2$ are analytical functions of $\gamma$, and mathematically we always have the inequality $\langle\hat{H}^2\rangle_\gamma\geq\langle\hat{H}\rangle_\gamma^2$, in the limit $\gamma\to 2$ where the equal sign happens, the leading order of the $(\gamma-2)$ expansion there should be as least the second order:

$$
\langle\hat{H}^2\rangle_\gamma-\langle\hat{H}\rangle_\gamma^2\propto(\gamma-2)^2\implies\tilde{E}_{\text{st}}^{(RR)}-\tilde{E}_{\text{st}}^{(W)}\propto|\gamma-2|.
\tag{A26}
$$

Therefore, it signals a sharp turn for $S_{\text{st}}^{(W)}$ at $\gamma=2$ due to the contribution from this non-differentiable absolute value function, as $S_{\text{st}}^{(RR)}$ is smooth there. This is indeed the behavior we observed.

*Appendix B.3. Application of the Wentzel–Kramers–Brillouin Approximation*

The Wentzel–Kramers–Brillouin approximation in $\mathcal{D} = 1$ uses semiclassical quantization conditions to estimate the eigenenergies via a summation series [69]:

$$\sum_{k=0}^{\infty} (-i)^{2k} \oint \Theta_{2k} = \sum_{k=0}^{\infty} = 2\pi \left( n + \frac{1}{2} \right) \quad \text{where} \quad n = \{0, 1, 2, 3, ...\} . \tag{A27}$$

For the ground state, we consider $n = 0$.

Perhaps the most familiar part of this general formula is the zeroth-order term, which is given by the total action in a single cycle of a classically allowed periodic trajectory:

$$\oint \Theta_0 = \oint dx \{ 2[\tilde{E} - \tilde{U}(x)] \}^{\frac{1}{2}} , \tag{A28}$$

which can be evaluated with $\tilde{U}(x) = |x|^{\gamma}$ to be

$$\oint \Theta_0 = 4\sqrt{2} \int_0^{+\tilde{E}^{\frac{1}{\gamma}}} dx \left[ \tilde{E} - |x|^{\gamma} \right]^{\frac{1}{2}} \xrightarrow{\rho = xE^{-\frac{1}{\gamma}}} 4\sqrt{2} \tilde{E}^{\frac{2+\gamma}{2\gamma}} \int_0^{+1} d\rho (1 - |\rho|^{\gamma})^{\frac{1}{2}}$$

$$= 4\sqrt{2} \tilde{E}^{\frac{2+\gamma}{2\gamma}} \left[ \frac{\sqrt{\pi}}{2} \frac{\Gamma\left(1 + \frac{1}{\gamma}\right)}{\Gamma\left(\frac{3}{2} + \frac{1}{\gamma}\right)} \right] = \sqrt{8\pi} \frac{\Gamma\left(1 + \frac{1}{\gamma}\right)}{\Gamma\left(\frac{3}{2} + \frac{1}{\gamma}\right)} \tilde{E}^{\frac{2+\gamma}{\gamma}} . \tag{A29}$$

Using this in Equation (A27), we can estimate the energy of the ground state:

$$\sqrt{8\pi} \frac{\Gamma\left(1 + \frac{1}{\gamma}\right)}{\Gamma\left(\frac{3}{2} + \frac{1}{\gamma}\right)} \left[ \tilde{E}_{\Omega}^{(\text{WKB},0)} \right]^{\frac{2+\gamma}{\gamma}} = 2\pi \left( n + \frac{1}{2} \right) \Big|_{n=0} = \pi$$

$$\implies \tilde{E}_{\Omega}^{(\text{WKB},0)} = \left[ \sqrt{\frac{\pi}{8}} \frac{\Gamma\left(\frac{3}{2} + \frac{1}{\gamma}\right)}{\Gamma\left(1 + \frac{1}{\gamma}\right)} \right]^{\frac{2\gamma}{2+\gamma}} . \tag{A30}$$

We can take this result and apply Equation (A4) to obtain a zeroth-order estimation for the stationary population success:

$$S_{\text{st}}^{(\text{WKB},0)} = 1 - \frac{2D}{R_0 \lambda^2} \left[ \tilde{E}_{\Omega}^{(\text{WKB},0)} \right]^{\frac{2+\gamma}{\gamma}} = 1 - \frac{\pi}{4} \frac{D}{R_0 \lambda^2} \left[ \frac{\Gamma\left(\frac{3}{2} + \frac{1}{\gamma}\right)}{\Gamma\left(1 + \frac{1}{\gamma}\right)} \right]^2 . \tag{A31}$$

For higher orders, the quantization conditions can be written as

$$\sum_{k=0}^{\infty} c_{2k} \mathcal{E}^{\frac{1}{2} - k} = 2\pi \left( n + \frac{1}{2} \right) \quad \text{where} \quad \mathcal{E} = \tilde{E}^{\frac{2+\gamma}{\gamma}} . \tag{A32}$$

The first few coefficients we calculated for the pure power-law potential $\tilde{U}(x) = |x|^{\gamma}$ to be

$$c_0 = 2\sqrt{2\pi} \frac{\Gamma\left(1 + \frac{1}{\gamma}\right)}{\Gamma\left(\frac{3}{2} + \frac{1}{\gamma}\right)} , \; c_2 = -\frac{\sqrt{2\pi}}{12} \frac{\gamma \Gamma\left(2 - \frac{1}{\gamma}\right)}{\Gamma\left(\frac{1}{2} - \frac{1}{\gamma}\right)} ,$$

$$c_4 = -\frac{\sqrt{2\pi}}{8640} \frac{\gamma^3 (3 + 2\gamma) \Gamma\left(4 - \frac{3}{\gamma}\right)}{(3 - 2\gamma) \Gamma\left(-\frac{1}{2} - \frac{3}{\gamma}\right)} . \tag{A33}$$

For tractability, we only go up the second order. Then, for the ground state, from Equation (A32), we can arrive at

$$c_0 \mathcal{E}_\Omega^{\frac{1}{2}} + c_2 \mathcal{E}_\Omega^{-\frac{1}{2}} = 2\pi \left( n + \frac{1}{2} \right) \bigg|_{n=0} = \pi \,, \tag{A34}$$

which can be solved as a quadratic polynomial,

$$\mathcal{E}_\Omega = \frac{\pi^2 - 2c_0 c_2 + \pi \sqrt{\pi^2 - 4c_0 c_2}}{2c_0^2} \,. \tag{A35}$$

Hence, by using Equation (A4), we can obtain the second-order estimation

$$S_{\text{st}}^{(\text{WKB},2)} = 1 - \frac{2D}{R_0 \lambda^2} \left( \frac{\pi^2 - 2c_0 c_2 + \pi \sqrt{\pi^2 - 4c_0 c_2}}{2c_0^2} \right) \,. \tag{A36}$$

**Appendix C. Simulation of the Non-Homogeneous Random Walk on the Landscape**

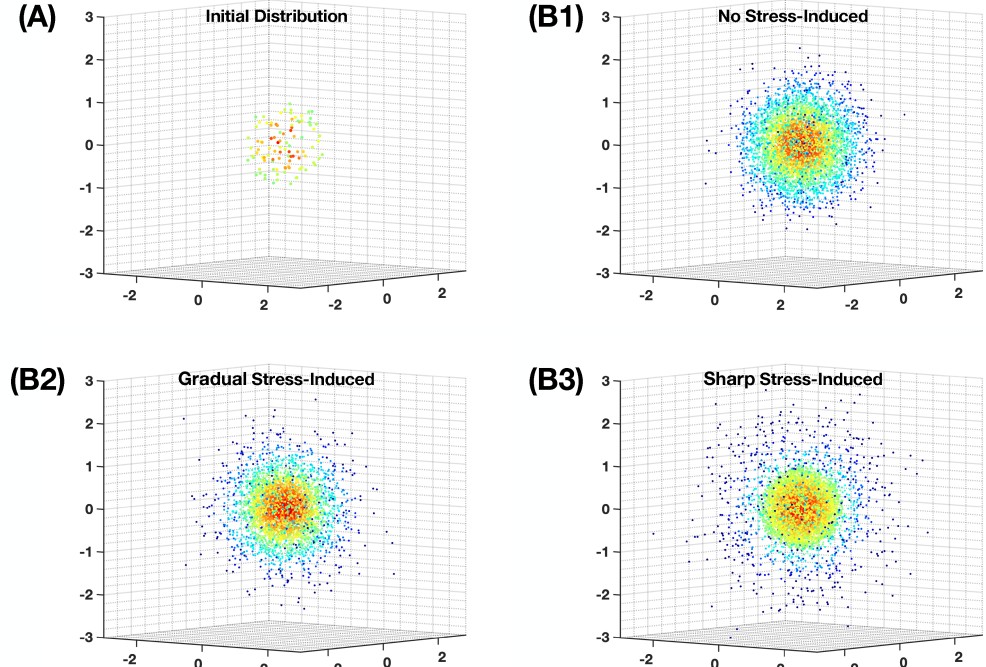

**Figure A1. Our simulation for the evolution of heterogeneous population distribution on the $\mathscr{D} = 3$-dimensional fitness landscape as described in Equation (10).** We use parameter values $D = 1/18$, $R_0 = 1$, $\lambda = 1$, and $K = 10^5$. For better visualization, only 10% of the agents are shown. (**A**) The initial distribution at $t = 0$ we use for all runs, using $\int d^3 \vec{x} b(\vec{x}, 0) = 10^3$ agents. (**B1**) A snapshot of the distribution at a stationary state if the evolution has no stress-induced mutagenesis. (**B2**) A snapshot of the distribution at a stationary state if the evolution has gradual stress-induced mutagenesis as in Equation (21), where $\epsilon = 0.1$. (**B3**) A snapshot of the distribution at a stationary state if the evolution has sharp stress-induced mutagenesis as in Equation (28), where $\epsilon = 10$.

We use agent-based simulations to investigate the population dynamics, in which each agent is specified by its location $\vec{x} = (x_1, x_2, ..., x_{\mathscr{D}})$ on the $\mathscr{D}$-dimensional landscape. We discretize the time $t$ into evenly pacing simulation time-steps, so that two consecutive steps are $\Delta t$ apart. At every simulation step, the position of each agent in every different direction $j \in \{1, 2, 3, ..., \mathscr{D}\}$ is updated with

$$x_j(t + \Delta t) = x_j(t) + \{2D[R(\vec{x})]\Delta t \times \mathcal{N}(0, 1)\} \,, \tag{A37}$$

where $D[R]$ is the fitness $R(\vec{x})$-dependence diffusivity, and $\mathcal{N}(0,1)$ is sample values from a Gaussian distribution of mean value 0 and standard deviation 1. Each agent also has a chance to multiply (when one agent becomes two) or die. These two are controlled by a single value $p$:

$$p = R(\vec{x})\left[1 - \frac{N(t)}{K}\right]\Delta t,\qquad\text{(A38)}$$

in which $K$ is the carrying capacity and $N(t)$ is the total number of agents at physical time $t$. A random number is generated uniformly between $[0,1]$, and if that number is larger than $|p|$, then the agent multiplies if $p > 0$ and dies if $p < 0$ (otherwise, nothing happens).

For fitness function $R(\vec{x})$, unless further specified, we use Equation (10). For the gradual stress-induced mutagenesis regime, we use $D[R]$ as in Equation (21). For the sharp stress-induced mutagenesis regime, we use $D[R]$ as in Equation (28). At $t = 0$, we use the very same initial distribution of $10^3$ agents in the landscape, which we place randomly using a uniformly generated procedure inside a ball of radius $\lambda$ (the *fit region* on the landscape).

For our simulations, we use a time discretization $\Delta = 0.01$ and the total time of $T = 100$. In every simulation, the population reaches a stationary state before $t = 50$, so all the population distribution densities are temporal averages of all agent position data in $t \in [51, 100]$. The values of other parameters ($\mathcal{D}, D, R_0, \lambda, K, \epsilon$) in different simulations are mentioned in the figures that show their results.

## Appendix D. Perturbative Corrections

*Appendix D.1. With Perturbed Hamiltonian Contains $|\vec{x}|^2$*

Following Equation (20), with perturbed Hamiltonian Equation (23), we can estimate the ground state energy shift via brute force integration as follows:

$$
\begin{aligned}
\delta E_{\Omega}^{(1)} &= \frac{\int_0^\infty \mathrm{d}|\vec{x}|\,|\vec{x}|^{\mathcal{D}-1}e^{-\frac{1}{2}\omega|\vec{x}|^2}\cdot\frac{1}{2}\hat{p}^2\left(\frac{|\vec{x}|}{\lambda}\right)^2\cdot e^{-\frac{1}{2}\omega|\vec{x}|^2}}{\int_0^\infty \mathrm{d}|\vec{x}|\,|\vec{x}|^{\mathcal{D}-1}e^{-\frac{1}{2}\omega|\vec{x}|^2}\cdot e^{-\frac{1}{2}\omega|\vec{x}|^2}}\\[2mm]
&= \frac{\int_0^\infty \mathrm{d}|\vec{x}|\,|\vec{x}|^{\mathcal{D}-1}e^{-\frac{1}{2}\omega|\vec{x}|^2}\cdot\frac{1}{2}(-\nabla^2)\left[\left(\frac{|\vec{x}|}{\lambda}\right)^2 e^{-\frac{1}{2}\omega|\vec{x}|^2}\right]}{\int_0^\infty \mathrm{d}|\vec{x}|\,|\vec{x}|^{\mathcal{D}-1}e^{-\omega|\vec{x}|^2}}\\[2mm]
&= \frac{-\frac{1}{2}\int_0^\infty \mathrm{d}|\vec{x}|\,|\vec{x}|^{\mathcal{D}-1}e^{-\omega|\vec{x}|^2}\left[\frac{\omega^2|\vec{x}|^4 - (\mathcal{D}+4)\omega|\vec{x}|^2 + 2\mathcal{D}}{\lambda^2}\right]}{\frac{1}{2}\omega^{-\frac{\mathcal{D}}{2}}\Gamma\left(\frac{\mathcal{D}}{2}\right)}\\[2mm]
&= \frac{-\dfrac{(\mathcal{D}-2)\omega^{-\frac{\mathcal{D}}{2}}\Gamma\left(1+\frac{\mathcal{D}}{2}\right)}{8\lambda^2}}{\frac{1}{2}\omega^{-\frac{\mathcal{D}}{2}}\Gamma\left(\frac{\mathcal{D}}{2}\right)} = \frac{\mathcal{D}(\mathcal{D}-2)}{8\lambda^2}.
\end{aligned}\qquad\text{(A39)}
$$

*Appendix D.2. With Perturbed Hamiltonian Contains $x^\kappa$*

In order to evaluate $\delta E_\Omega^{(1)}$ for the $\hat{p}^2|\vec{x}|^\kappa$ perturbation operator, we start from Equation (20):

$$
\begin{aligned}
\delta E_\Omega^{(1)} &= \frac{\int_0^\infty \mathrm{d}|\vec{x}||\vec{x}|^{\mathscr{D}-1}e^{-\frac{1}{2}\omega|\vec{x}|^2}\cdot\frac{1}{2}\hat{p}^2\left(\frac{|\vec{x}|}{\lambda}\right)^\kappa\cdot e^{-\frac{1}{2}\omega|\vec{x}|^2}}{\int_0^\infty \mathrm{d}|\vec{x}||\vec{x}|^{\mathscr{D}-1}e^{-\frac{1}{2}\omega|\vec{x}|^2}\cdot e^{-\frac{1}{2}\omega|\vec{x}|^2}} \\[2mm]
&= \frac{\int_0^\infty \mathrm{d}|\vec{x}||\vec{x}|^{\mathscr{D}-1}e^{-\frac{1}{2}\omega|\vec{x}|^2}\cdot\frac{1}{2}(-\nabla^2)\left[\left(\frac{|\vec{x}|}{\lambda}\right)^\kappa e^{-\frac{1}{2}\omega|\vec{x}|^2}\right]}{\int_0^\infty \mathrm{d}|\vec{x}||\vec{x}|^{\mathscr{D}-1}e^{-\omega|\vec{x}|^2}} \\[2mm]
&= \frac{-\frac{1}{2}\int_0^\infty \mathrm{d}|\vec{x}||\vec{x}|^{\mathscr{D}+\kappa-1}e^{-\omega|\vec{x}|^2}\left[\dfrac{\omega^2|\vec{x}|^4-(\mathscr{D}+2\kappa)\omega|\vec{x}|^2+\kappa(\mathscr{D}+\kappa-2)}{\lambda^\kappa}\right]}{\frac{1}{2}\omega^{-\frac{\mathscr{D}}{2}}\Gamma\left(\dfrac{\mathscr{D}}{2}\right)} \\[2mm]
&= \frac{-\dfrac{(\mathscr{D}-\kappa)\omega^{1-\frac{\mathscr{D}+\kappa}{2}}\Gamma\left(\frac{\mathscr{D}+\kappa}{2}\right)}{8\lambda^\kappa}}{\frac{1}{2}\omega^{-\frac{\mathscr{D}}{2}}\Gamma\left(\dfrac{\mathscr{D}}{2}\right)} = \frac{(\mathscr{D}-\kappa)\omega^{1-\frac{\kappa}{2}}\Gamma\left(\dfrac{\mathscr{D}+\kappa}{2}\right)}{4\lambda^\kappa\Gamma\left(\dfrac{\mathscr{D}}{2}\right)}.
\end{aligned}
\tag{A40}
$$

For the sanity check, when we substitute $\kappa = 2$, result $\delta E_\Omega^{(1)}$ becomes Equation (A39).

*Appendix D.3. With Perturbed Hamiltonian Contains Heaviside Function*

In order to evaluate $\delta E_\Omega^{(1)}$ for the non-Hermitian operator containing a Heaviside function, which is given in Equation (30), we expand Equation (20):

$$
\begin{aligned}
\delta E_\Omega^{(1)} &= \frac{\int_0^\infty \mathrm{d}|\vec{x}||\vec{x}|^{\mathscr{D}-1}e^{-\frac{1}{2}\omega|\vec{x}|^2}\cdot\frac{1}{2}\hat{p}^2\Theta\left(\dfrac{|\vec{x}|}{\lambda}-1\right)\cdot e^{-\frac{1}{2}\omega|\vec{x}|^2}}{\int_0^\infty \mathrm{d}|\vec{x}||\vec{x}|^{\mathscr{D}-1}e^{-\frac{1}{2}\omega|\vec{x}|^2}\cdot e^{-\frac{1}{2}\omega|\vec{x}|^2}} \\[2mm]
&= \frac{\int_0^\infty \mathrm{d}|\vec{x}||\vec{x}|^{\mathscr{D}-1}e^{-\frac{1}{2}\omega|\vec{x}|^2}\cdot\frac{1}{2}(-\nabla^2)\left[\Theta\left(\dfrac{|\vec{x}|}{\lambda}-1\right)e^{-\frac{1}{2}\omega|\vec{x}|^2}\right]}{\int_0^\infty \mathrm{d}|\vec{x}||\vec{x}|^{\mathscr{D}-1}e^{-\omega|\vec{x}|^2}} \\[2mm]
&= \frac{-\frac{1}{2}\int_0^\infty \mathrm{d}|\vec{x}|e^{-\frac{1}{2}\omega|\vec{x}|^2}\cdot\partial_{|\vec{x}|}\left\{|\vec{x}|^{\mathscr{D}-1}\partial_{|\vec{x}|}\left[\Theta\left(\dfrac{|\vec{x}|}{\lambda}-1\right)e^{-\frac{1}{2}\omega|\vec{x}|^2}\right]\right\}}{\frac{1}{2}\omega^{-\frac{\mathscr{D}}{2}}\Gamma\left(\dfrac{\mathscr{D}}{2}\right)}.
\end{aligned}
\tag{A41}
$$

It is non-trivial to deal with the derivatives of the Heaviside function, so let us move slower:

$$
\begin{aligned}
\partial_{|\vec{x}|}&\left\{|\vec{x}|^{\mathscr{D}-1}\partial_{|\vec{x}|}\left[\Theta\left(\frac{|\vec{x}|}{\lambda}-1\right)e^{-\frac{1}{2}\omega|\vec{x}|^2}\right]\right\}\\
&=\partial_{|\vec{x}|}\left\{|\vec{x}|^{\mathscr{D}-1}e^{-\frac{1}{2}\omega|\vec{x}|^2}\left[-\omega|\vec{x}|\Theta\left(\frac{|\vec{x}|}{\lambda}-1\right)+\frac{1}{\lambda}\delta\left(\frac{|\vec{x}|}{\lambda}-1\right)\right]\right\}\\
&=\omega\left(\omega|\vec{x}|^2-\mathscr{D}\right)x^{\mathscr{D}-1}e^{-\frac{1}{2}\omega|\vec{x}|^2}\Theta\left(\frac{|\vec{x}|}{\lambda}-1\right)\\
&\quad+\frac{1}{\lambda}\left[-2\omega|\vec{x}|^{\mathscr{D}}+(\mathscr{D}-1)|\vec{x}|^{\mathscr{D}-2}\right]e^{-\frac{1}{2}\omega|\vec{x}|^2}\delta\left(\frac{|\vec{x}|}{\lambda}-1\right)\\
&\quad+\frac{1}{\lambda}|\vec{x}|^{\mathscr{D}-1}e^{-\frac{1}{2}\omega|\vec{x}|^2}\partial_{|\vec{x}|}\delta\left(\frac{|\vec{x}|}{\lambda}-1\right),
\end{aligned}
\tag{A42}
$$

where $\delta(...)$ is the Dirac-delta function [89]. To proceed, we evaluate the integration of each term separately. The $\Theta$-term:

$$
\begin{aligned}
-\frac{1}{2}&\int_0^\infty d|\vec{x}|e^{-\frac{1}{2}\omega|\vec{x}|^2}\left[\omega\left(\omega|\vec{x}|^2-\mathscr{D}\right)x^{\mathscr{D}-1}e^{-\frac{1}{2}\omega|\vec{x}|^2}\Theta\left(\frac{|\vec{x}|}{\lambda}-1\right)\right]\\
&=-\frac{1}{2}\int_\lambda^\infty d|\vec{x}|x^{\mathscr{D}-1}e^{-\omega|\vec{x}|^2}\omega\left(\omega|\vec{x}|^2-\mathscr{D}\right)\\
&=\frac{1}{4}\omega\lambda^{\mathscr{D}}\left[-2e^{-\omega\lambda^2}+\omega\lambda^2\,\mathrm{E}_{-\frac{\mathscr{D}}{2}}\left(\omega\lambda^2\right)\right],
\end{aligned}
\tag{A43}
$$

the $\delta$-term:

$$
\begin{aligned}
-\frac{1}{2}&\int_0^\infty d|\vec{x}|e^{-\frac{1}{2}\omega|\vec{x}|^2}\left\{\frac{1}{\lambda}\left[-2\omega|\vec{x}|^{\mathscr{D}}+(\mathscr{D}-1)|\vec{x}|^{\mathscr{D}-2}\right]e^{-\frac{1}{2}\omega|\vec{x}|^2}\delta\left(\frac{|\vec{x}|}{\lambda}-1\right)\right\}\\
&=-\frac{1}{2\lambda}\left[-2\omega|\vec{x}|^{\mathscr{D}}+(\mathscr{D}-1)|\vec{x}|^{\mathscr{D}-2}\right]e^{-\omega|\vec{x}|^2}\bigg|_{|\vec{x}|=\lambda}\\
&=-\frac{1}{2\lambda}\left[-2\omega\lambda^{\mathscr{D}}+(\mathscr{D}-1)\lambda^{\mathscr{D}-2}\right]e^{-\omega\lambda^2},
\end{aligned}
\tag{A44}
$$

and finally the $\partial_{|\vec{x}|}\delta$-term:

$$
\begin{aligned}
-\frac{1}{2}&\int_0^\infty d|\vec{x}|e^{-\frac{1}{2}\omega|\vec{x}|^2}\left[\frac{1}{\lambda}|\vec{x}|^{\mathscr{D}-1}e^{-\frac{1}{2}\omega|\vec{x}|^2}\partial_{|\vec{x}|}\delta\left(\frac{|\vec{x}|}{\lambda}-1\right)\right]\\
&=-\frac{1}{2\lambda}\int_0^\infty d|\vec{x}||\vec{x}|^{\mathscr{D}-1}e^{-\omega|\vec{x}|^2}\partial_{|\vec{x}|}\delta\left(\frac{|\vec{x}|}{\lambda}-1\right)\\
&=\frac{1}{2\lambda}\int_0^\infty d|\vec{x}|\partial_{|\vec{x}|}\left(|\vec{x}|^{\mathscr{D}-1}e^{-\omega|\vec{x}|^2}\right)\delta\left(\frac{|\vec{x}|}{\lambda}-1\right)\\
&=\frac{1}{2\lambda}|\vec{x}|^{\mathscr{D}-2}\left(\mathscr{D}-2\omega|\vec{x}|^2-1\right)e^{-\omega|\vec{x}|^2}\bigg|_{|\vec{x}|=\lambda}\\
&=\frac{1}{2\lambda}\left[-2\omega\lambda^{\mathscr{D}}+(\mathscr{D}-1)\lambda^{\mathscr{D}-2}\right]e^{-\omega\lambda^2},
\end{aligned}
\tag{A45}
$$

in which we used integration by part. Adding up these three, we obtain the numerator of Equation (A41), which provides

$$
\begin{aligned}
\delta E_\Omega^{(1)} &= \frac{\frac{1}{4}\omega\lambda^{\mathscr{D}}\left[-2e^{-\omega\lambda^2} + \omega\lambda^2\,\mathrm{E}_{-\frac{\mathscr{D}}{2}}\left(\omega\lambda^2\right)\right]}{\frac{1}{2}\omega^{-\frac{\mathscr{D}}{2}}\Gamma\left(\frac{\mathscr{D}}{2}\right)} \\[2mm]
&= \frac{1}{\lambda^2}\frac{(\omega\lambda^2)^{\frac{\mathscr{D}}{2}+1}\left[-2e^{-\omega\lambda^2} + \omega\lambda^2\,\mathrm{E}_{-\frac{\mathscr{D}}{2}}\left(\omega\lambda^2\right)\right]}{2\Gamma\left(\frac{\mathscr{D}}{2}\right)}\ .
\end{aligned}
\tag{A46}
$$

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
