# Peer review of "A Schrödinger Equation for Evolutionary Dynamics"

_quantumrep, doi:10.3390/quantum5040042_

Round 1

Reviewer 1 Report

Comments and Suggestions for Authors

This is a creative work applying quantum mechanics to biophysics. It can be inspirational for the quantum community to think about the quantum tools’ new application to other fields.

The work indeed uses some results from quantum mechanics, such as the energies in harmonic oscillator, and tools like perturbation theorem for the stresses mutagenesis to suggest some biologically meaningful possibility.

I recommend this paper to be published in Quantum Reports. Nonetheless, there are typos can be further checked, such as the following:

Line 149: zero-eigenenery -> zero-eigenenergy

Line 191: Schordinger -> Schrödinger

Comments on the Quality of English Language

Please see above regarding the typo.

Author Response

Please see the attachment, section Referee Report 1.

Reviewer 2 Report

Comments and Suggestions for Authors

Review Comments

Manuscript ID: quantumrep-2558329

 I have gone through the whole manuscript titled onA Schrodinger-Bloch Equation for Evolutionary Dynamics”.

Overall, the presentation of the article looks very good and English grammar is also good.  All the terms and formulas are well defined. My decision is, the manuscript is potentially publishable after minor revisions.

1.     The authors have claimed the use of two different methods; as in line 44, Rayleigh-Ritz variational method and in line 66, Rayleigh-Schordinger perturbation theory and find the solution. The authors are requested to give some comparison with the existing methods for novelty of their work.

2.     The discussion section must be more specific about the findings.

Author Response

Please see the attachment, section Referee Report 2.

Reviewer 3 Report

Comments and Suggestions for Authors

Please see the full review attached.

Comments on the Quality of English Language

Minor spellchecks are required.

Author Response

Please see the attachment, section Referee Report 3.
